# The molecular architecture of engulfment during *Bacillus subtilis* sporulation

**Kanika Khanna, Javier Lopez-Garrido, Ziyi Zhao, Reika Watanabe, Yuan Yuan, Joseph Sugie, Kit Pogliano\*, Elizabeth Villa\***

Division of Biological Sciences, University of California, San Diego, La Jolla, United States

**Abstract** The study of bacterial cell biology is limited by difficulties in visualizing cellular structures at high spatial resolution within their native milieu. Here, we visualize *Bacillus subtilis* sporulation using cryo-electron tomography coupled with cryo-focused ion beam milling, allowing the reconstruction of native-state cellular sections at molecular resolution. During sporulation, an asymmetrically-positioned septum generates a larger mother cell and a smaller forespore. Subsequently, the mother cell engulfs the forespore. We show that the septal peptidoglycan is not completely degraded at the onset of engulfment. Instead, the septum is uniformly and only slightly thinned as it curves towards the mother cell. Then, the mother cell membrane migrates around the forespore in tiny finger-like projections, whose formation requires the mother cell SpoIIDMP protein complex. We propose that a limited number of SpoIIDMP complexes tether to and degrade the peptidoglycan ahead of the engulfing membrane, generating an irregular membrane front.

DOI: https://doi.org/10.7554/eLife.45257.001

## Introduction

From an architectural point of view, bacterial cells are among the simplest forms of life on the planet. Their cytoplasm is typically devoid of membrane bound organelles, and their cellular morphology relies upon a semi-rigid peptidoglycan (PG) cell wall that imposes its shape on the malleable cell membrane(s). Bacterial cells are inflated by their high internal turgor pressure, which pushes the membranes against the cell wall, causing the PG to stretch and the cell to adopt its appropriate shape. Despite the apparent simplicity, studies in the past few decades have demonstrated that bacterial cellular architecture is far more complex than previously thought, in terms of both its ultrastructure and dynamic capabilities (*Hawver et al., 2016*; *Wagstaff and Löwe, 2018*; *Wang et al., 2013*).

Endospore formation in *Bacillus subtilis* represents a striking example of the dynamic capabilities of bacterial cells, as it entails dramatic changes in cellular architecture. First, the division site shifts to polar position, generating a sporangium comprised of two cells: a larger mother cell and a smaller forespore (*Figure 1A*; *Errington, 2003*; *Higgins and Dworkin, 2012*; *Tan and Ramamurthi, 2014*). The polar septum traps the forespore chromosome, which is subsequently transported to the forespore by SpoIIIE, a membrane-anchored ATPase that assembles a translocation complex at septal midpoint (*Bath et al., 2000*; *Yen Shin et al., 2015*; *Wu and Errington, 1994*; *Wu and Errington, 1997*). Chromosome translocation increases the turgor pressure in the forespore, causing it to inflate and expand into the mother cell (*Lopez-Garrido et al., 2018*). Simultaneously, the mother cell engulfs the forespore in a process that resembles eukaryotic phagocytosis (*Figure 1A*). After engulfment, the forespore is fully enclosed within the cytoplasm of the mother cell, where it matures in a process that involves the synthesis of protective layers of cortex and coat, and the partial dehydration of the forespore cytoplasm. Finally, the mother cell lyses and the mature spore is released to the environment, where it remains dormant until the conditions are appropriate for germination.

**\*For correspondence:**
kpogliano@ucsd.edu (KP);
evilla@ucsd.edu (EV)

**Competing interests:** The authors declare that no competing interests exist.

**eLife digest** Much of what happens in biology occurs at scales so small that the microscopy methods traditionally used by biologists cannot visualize them. One such process is bacterial sporulation: in stressful conditions, bacteria like *Bacillus subtilis* can divide to produce a smaller cell called a forespore, which the larger mother cell then engulfs. The forespore matures into a hardy spore, which is able to survive in harsh environments and only transform into an active bacterium when conditions improve.

Bacterial cells are surrounded by a stiff layer of a material called peptidoglycan. This wall sits outside of the bacterium's thin flexible membrane and determines the bacterium's shape. At the beginning of sporulation, the forespore is separated from the mother cell by a peptidoglycan wall. Engulfment of the spore by the mother cell requires a dramatic change in the shape of this partition. Microbiologists had thought that all the rigid peptidoglycan must be degraded to allow the partition to deform flexibly during engulfment; however, no one had yet observed the tiny structures involved.

Khanna et al. directly visualized sporulation in *B. subtilis* using a technique called cryo-electron tomography (or cryo-ET for short). In cryo-ET, samples are cooled to low temperatures and then imaged with a beam of electrons. Cryo-ET requires thin samples, thinner even than most bacteria. By combing cryo-ET with another methodology that allowed them to focus in on thin sections of their sample, Khanna et al. generated high resolution images, which provided a look at forespore engulfment in unprecedented detail.

These images revealed that the peptidoglycan wall separating the mother cell from forespore is not completely degraded: a thin layer of peptidoglycan persists. Comparing these images to cryo-ET images of cells treated with drugs that block the production of peptidoglycan suggested a new engulfment mechanism. This includes a cycle of peptidoglycan production and degradation that accompanies the advancing mother cell membrane as it surrounds the forespore during engulfment. Khanna et al. could also see that the mother cell's membrane formed finger-like projections as it moved around the forespore, something that was not visible with previous techniques.

This detailed engulfment mechanism is an important advance in the understanding of bacterial spore formation. Additionally, Khanna et al. have generated a collection of images, methods and analyses that may prove useful to a wide community of biologists attempting to understand sporulation and other fundamental processes that occur on a small scale.

DOI: https://doi.org/10.7554/eLife.45257.002

Engulfment represents a major rearrangement of the sporangium, from two cells that lie side by side to a cell within a cell. Such rearrangement likely involves a profound remodeling of the PG cell wall around the forespore, which would otherwise constrain the movement of the engulfing membrane. At the onset of engulfment, the engulfing mother cell membrane must circumvent the physical barrier posed by the septal PG in order to migrate around the forespore. This has led to the logical proposal that engulfment must entail the complete dissolution of the septal PG, a process often referred to as 'septal thinning' (*Chastanet and Losick, 2007*; *Illing and Errington, 1991*; *Perez et al., 2000*). This proposal was supported by early electron microscopy studies of fixed and stained sporangia showing that engulfment-defective mutants had thicker septa than wild type sporangia (*Holt et al., 1975*; *Illing and Errington, 1991*; *Rousseau and Hermier, 1975*). Further studies showed that engulfment requires three mother cell proteins: SpoIID, SpoIIM and SpoIIP, which form a complex (DMP) with PG degradation activity (*Abanes-De Mello et al., 2002*; *Aung et al., 2007*; *Chastanet and Losick, 2007*; *Morlot et al., 2010*). In principle, DMP could mediate the complete dissolution of the septal PG to remove the steric block to the movement of the mother cell membrane around the forespore.

The idea that septal PG is completely degraded has been more recently challenged by cryo-electron tomography (cryo-ET) images showing that a thin PG layer is present between the forespore and the mother cell membranes throughout engulfment (*Tocheva et al., 2013*). It has also been shown that DMP-mediated PG degradation is required and rate-limiting for membrane migration even after the septal barrier has been bypassed, suggesting that DMP plays a role separate from the

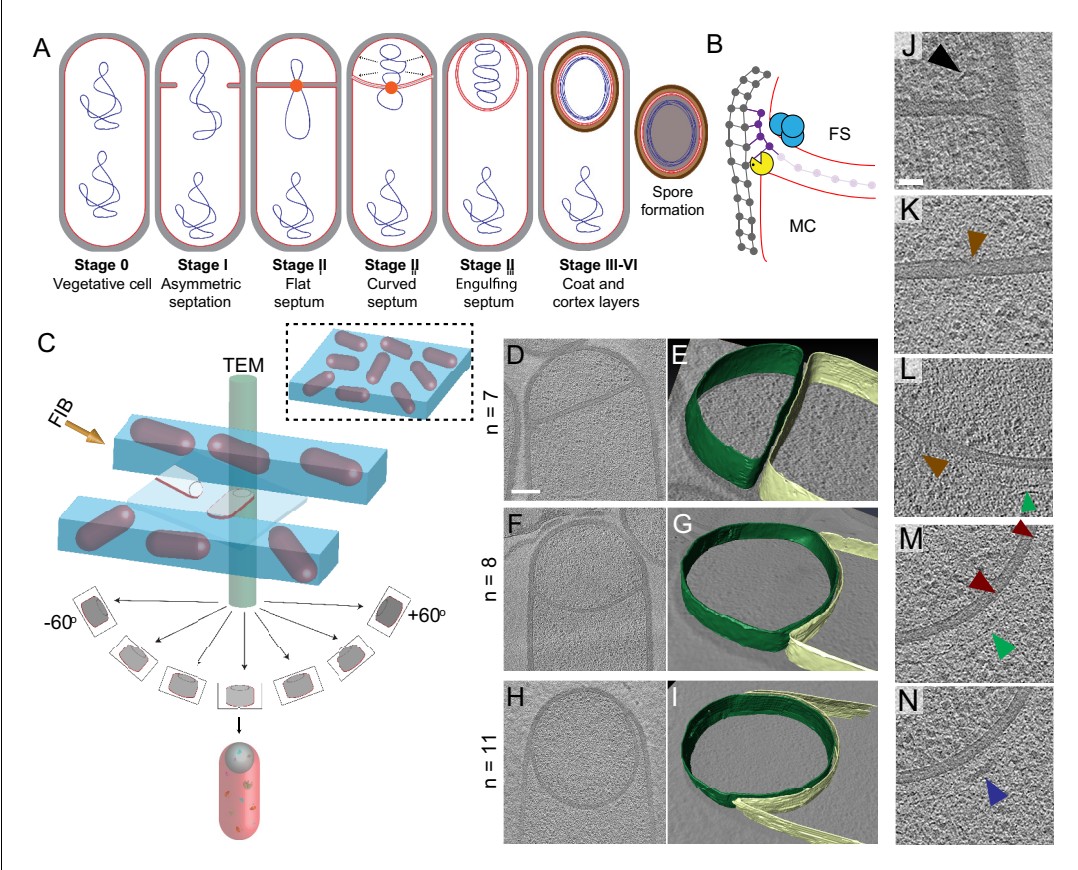

**Figure 1.** Visualizing the 3D architecture of engulfment during sporulation in *B. subtilis*. (**A**) Schematic illustrating the process of polar septation, chromosome translocation, engulfment and spore maturation. Membranes (red), PG (gray), chromosomes (blue), SpoIIIE (orange) and coat proteins (shades of brown) are highlighted. Outward arrows in the stage II$_{ii}$ forespore indicate increased turgor pressure due to chromosome translocation. (**B**) Revised engulfment model (*Ojkic et al., 2016*): new PG (purple) is synthesized ahead of the leading edge of the engulfing membrane by forespore-associated PG biosynthetic enzymes (blue) and is subsequently degraded by DMP (yellow pacman), making room for the engulfing membrane to advance. The coordinated synthesis and degradation of PG at the leading edge of the engulfing membrane can move the junction between the septum (pink) and the lateral cell wall (gray) around the forespore. (**C**) Schematic illustrating cryo-FIB-ET methodology for bacterial samples (see Materials and methods). (**D–I**) Slices through cryo-electron tomograms representing different stages of engulfment: (**D**) flat polar septum (Stage II$_i$), (**F**) curved septum (Stage II$_{ii}$) and (**H**) engulfing septum (Stage II$_{iii}$). Scale bar for (D,F,I): 200 nm. The corresponding forespore membrane (green) and the mother cell membrane (yellow) are annotated in (**E,G,I**) respectively. n indicates the number of tomograms acquired for each cell type. Scale bars have been omitted for (E,G,I) as cells are shown in perspective views . (**J–N**) Zoomed-in slices through cryo-electron tomograms showing (**J**) a large ellipsoidal complex adjacent to the forespore membrane (black arrow), (**K**) a putative SpoIIIE channel (brown arrow) and (**L**) another putative channel (brown arrow), (**L,M**) coat filaments (green arrows), (**M**) basement coat layer (maroon arrow) and (**N**) amorphous coat proteins (purple arrow). Scale bar for (J-N): 50 nm.

DOI: https://doi.org/10.7554/eLife.45257.003

The following figure supplements are available for figure 1:

**Figure supplement 1.** Slices through cryo-electron tomograms representing different stages of engulfment in wild type sporulating *B. subtilis* cells.

DOI: https://doi.org/10.7554/eLife.45257.004

**Figure supplement 2.** Slices through cryo-electron tomograms representing later stages of sporulation when the forespore is completely inside the mother cell.

DOI: https://doi.org/10.7554/eLife.45257.005

**Figure supplement 3.** Unidentified complexes during sporulation in *B. subtilis*.

DOI: https://doi.org/10.7554/eLife.45257.006

dissolution of septal PG (*Abanes-De Mello et al., 2002*; *Gutierrez et al., 2010*). In addition, the movement of the mother cell membrane also requires PG synthesis at the leading edge of the engulfing membrane (*Meyer et al., 2010*; *Ojkic et al., 2016*). Based on these observations, we proposed a revised model for engulfment membrane migration in which coordinated PG synthesis and

degradation at the leading edge of the engulfing mother cell membrane moves the junction between the septum and the lateral cell wall around the forespore, making room for the engulfing membrane to advance (*Figure 1B*; *Ojkic et al., 2016*). This model eliminates the need for complete dissolution of the septal PG and predicts that PG synthesis happens ahead of the leading edge of the engulfing membrane. Then, the mother cell DMP degrades this new PG to mediate membrane migration. However, due to the limited resolution of optical microscopy, conclusive evidence that PG synthesis occurs ahead of PG degradation is lacking.

Cryo-ET allows the visualization of three-dimensional (3D) architecture of bacterial membranes and cell wall in a hydrated near-native state that cannot be achieved by methods reliant on chemical fixation and staining (*Ben-Harush et al., 2010*; *Matias and Beveridge, 2005*; *Oikonomou et al., 2016*). However, a limitation of cryo-ET is that the samples must be less than ~500 nm thick to obtain high-resolution tomograms, constraining its application to only a handful of bacteria that are either naturally slender, or, as in the case of *B. subtilis*, for which slender mutant strains are available (*Tocheva et al., 2013*). But the latter typically contain mutations in genes involved in PG metabolism and may not be ideal to study cell wall remodeling. Recent application of cryo-focused ion beam (cryo-FIB) milling has produced artifact-free thin sample sections of ~100–300 nm, which allows the acquisition of high-resolution tomograms of sections of wild type cells (*Figure 1C*; *Marko et al., 2007*; *Rigort et al., 2012*; *Villa et al., 2013*). Cryo-FIB milling coupled with cryo-ET (or cryo-FIB-ET) is therefore becoming the method of choice for studies of cellular architecture of both eukaryotic and prokaryotic cells (*Chaikeeratisak et al., 2017*; *Engel et al., 2015*; *Lopez-Garrido et al., 2018*; *Mahamid et al., 2016*).

Here, we have used cryo-FIB-ET to study sporulation in *B. subtilis*, revealing the different stages of engulfment with a resolution that has not been achieved previously. We have analyzed wild type sporangia, engulfment mutants, and sporangia treated with PG synthesis inhibitors to obtain new mechanistic insights into the PG transformations that occur during engulfment. First, we provide evidence that septal PG is not degraded completely at the onset of engulfment. Second, we show that during membrane migration, the newly synthesized PG deforms the forespore membrane ahead of the leading edge of the engulfing mother cell membrane, indicating that PG synthesis precedes PG degradation. Third, we observe that the mother cell membrane migrates around the forespore by forming tiny finger-like projections, the formation of which depends on DMP complexes tethering to and degrading the PG ahead of the engulfing membrane. The methodology, images and analyses presented here will provide valuable resources for future studies of spore assembly and other fundamental cell biological processes.

## Results

### Visualizing sporulation in wild type *B. subtilis* at molecular resolution

Recently, we used cryo-FIB-ET to illustrate the role of DNA translocation in inflating the forespore (*Lopez-Garrido et al., 2018*). These data confirmed the presence of a thin layer of PG between the forespore and the mother cell membranes in the wild type strain, as previously visualized by cryo-ET of a slender *ponA* mutant of *B. subtilis* (*Tocheva et al., 2013*). We expanded our cryo-FIB-ET studies to investigate the architecture of *B. subtilis* cells during different stages of sporulation (*Figure 1C*; see Materials and methods). We acquired high-quality data of wild type cells during engulfment (Stage II, *Figure 1D–I*; *Figure 1—figure supplement 1*) and during later stages of sporulation, when cortex and coat were being assembled (Stages III-VI, *Figure 1—figure supplement 2*). Data of very late stages of

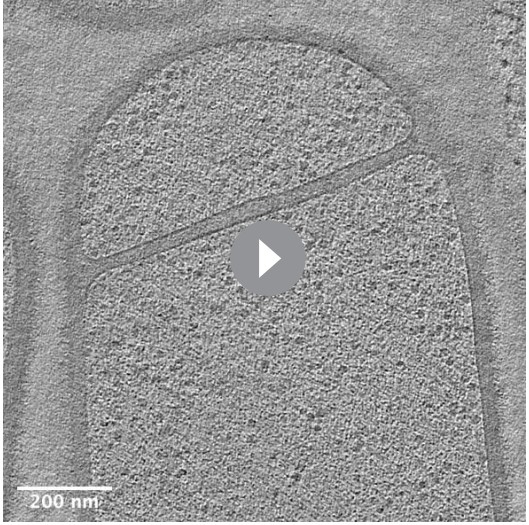

**Video 1.** Movie showing slices through a cryo-electron tomogram of *B. subtilis* wild type sporangium (flat septum, Stage II$_i$) shown in *Figure 1D*.
DOI: https://doi.org/10.7554/eLife.45257.007

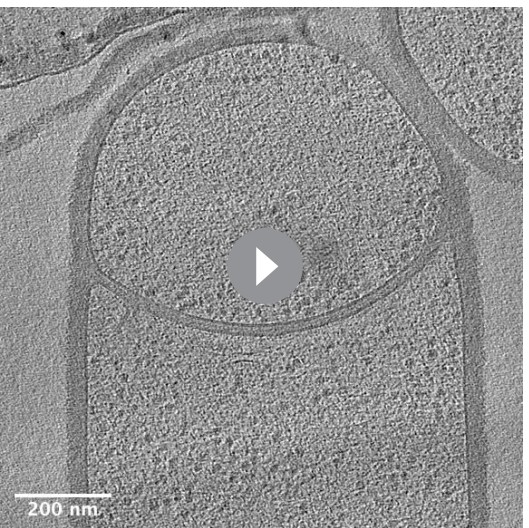

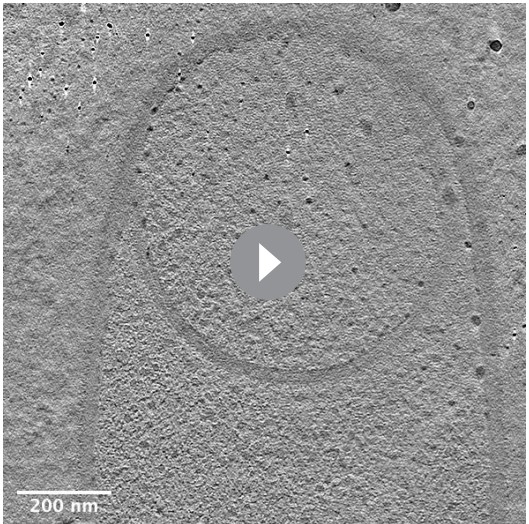

**Video 2.** Movie showing slices through a cryo-electron tomogram of *B. subtilis* wild type sporangium (curved septum, Stage II$_{ii}$) shown in **Figure 1F**.
DOI: https://doi.org/10.7554/eLife.45257.008

**Video 3.** Movie showing slices through a cryo-electron tomogram of *B. subtilis* wild type sporangium (engulfing septum, Stage II$_{iii}$) shown in **Figure 1H**.
DOI: https://doi.org/10.7554/eLife.45257.009

sporulation failed to provide high-resolution information inside the forespore, likely due to the dehydration of this cell, which increases sensitivity of cryo-ET samples to the electron beam (**Wu et al., 2012**).

Our data showed that the external cell wall of sporulating cells was ~20–30 nm thick, consistent with other EM studies (**Matias and Beveridge, 2005**; **Tocheva et al., 2013**). The polar septum formed close (within 500 nm) to one cell pole, and was initially flat (**Figure 1D,E**, **Figure 1—figure supplement 1A–D**, **Video 1**). Eventually, the septum bent smoothly into the mother cell (**Figure 1F, G**, **Figure 1—figure supplement 1E–H**, **Video 2**) and the mother cell membrane moved forward to engulf the forespore, at which stage the forespore was roughly rounded (**Figure 1H,I**, **Figure 1—figure supplement 1I–L**, **Video 3**).

Visual inspection of tomograms also revealed several structures that have not been characterized previously. Immediately after polar septation, we observed ellipsoidal complexes that were present only in the forespore, adjacent to the membrane and often close to the intersection between the septum and the lateral cell wall (**Figure 1J**, **Figure 1—figure supplement 3A–C**). These structures were observed in 2 out of 7 tomograms of wild type sporangia with flat septa, with ~4 ellipsoidal complexes observed in ~200 nm slices. The 3D reconstruction of these complexes revealed that the ellipsoidal structures had a mean radius of ~45 nm (**Figure 1—figure supplement 3A–C**, see Materials and methods). The molecular identity of these structures remains undetermined. We also identified a region approximately in the center of a flat septum where the two membranes are closer together than elsewhere on the septum (14 nm vs. 23 nm, **Figure 1K**, **Figure 1—figure supplement 3D,E**). This constriction may correspond to paired hexameric SpoIIIE channels (**Fleming et al., 2010**; **Liu et al., 2006**; **Yen Shin et al., 2015**). We also observed structures that appear to be channels crossing the septum (**Figure 1L**) that may correspond to SpoIIA-SpoIIQ complexes (**Blaylock et al., 2004**; **Levdikov et al., 2012**; **Morlot and Rodrigues, 2018**; **Zeytuni et al., 2017**). We observed this thin region in the polar septum and channel-like features (**Figure 1K,L**) in 2 out of 15 tomograms of wild type sporangia with flat and curved septa. The low frequency of observation could be attributed to the fact that we are imaging only ~150–250 nm slices of cells that are over 1 μm wide. So, many tomograms would not include low abundance structures such as SpoIIIE, which is comprised of two adjacent channels. Next, we observed a basement coat layer adjacent to the outer forespore membrane that is likely comprised of SpoVM and/or SpoIVA (**Figure 1M**; **Ramamurthi and Losick, 2009**). This layer is visible as an array of dots spaced ~4–6 nm apart, similar to that observed previously in *Acetonema longum* sporulating cells (**Tocheva et al., 2011**). Moving outward, we observed

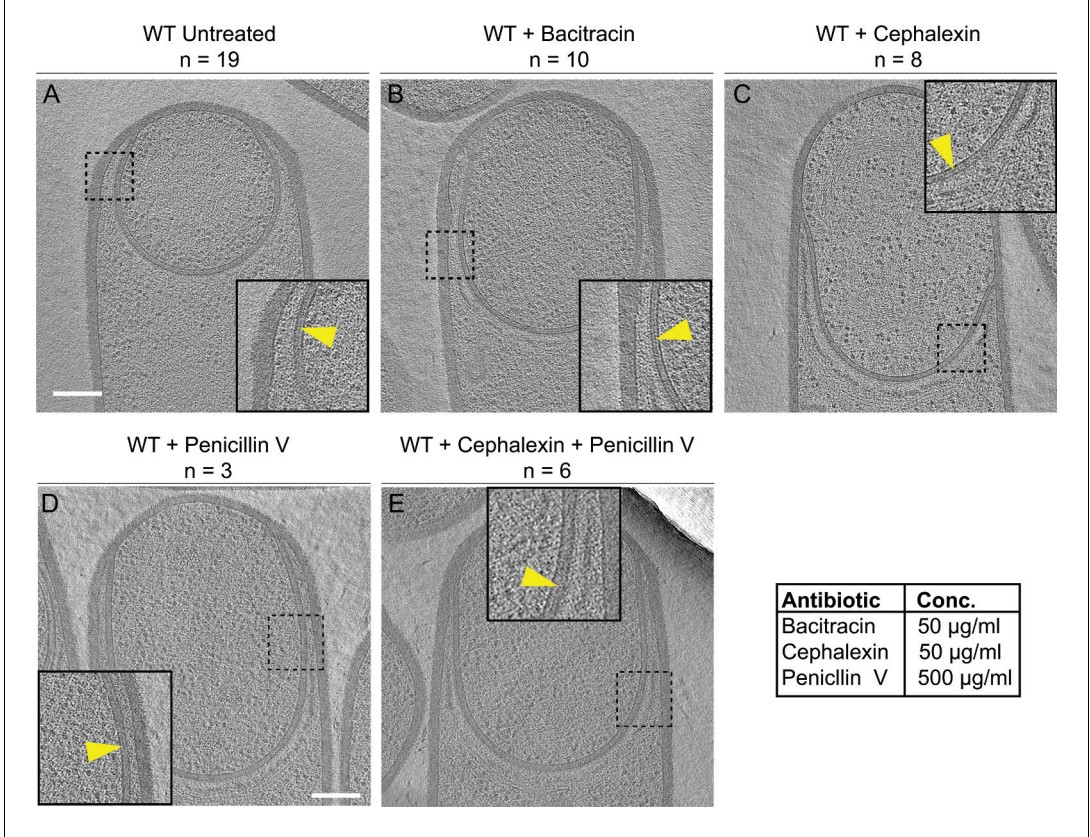

**Figure 2.** Septal PG is not completely degraded at the onset of engulfment. (**A–C**) Slices through cryo-electron tomograms of wild type engulfing sporangia that are (**A**) untreated, (**B**) bacitracin-treated, (**C**) cephalexin-treated, (**D**) penicillin V-treated and (**E**) penicillin V- and cephalexin-treated. The thin layer of septal PG is indicated by yellow arrows in the zoom-in panels for each of the tomographic slices. n indicates the number of tomograms acquired for each cell type. Antibiotic concentrations used for the experiments are indicated in the bottom right. Scale bar: 200 nm.

DOI: https://doi.org/10.7554/eLife.45257.010

The following figure supplements are available for figure 2:

**Figure supplement 1.** Quantification of cell length upon antibiotic treatment.

DOI: https://doi.org/10.7554/eLife.45257.011

**Figure supplement 2.** Thin layer of septal peptidoglycan persists in wild type and antibiotic-treated sporangia.

DOI: https://doi.org/10.7554/eLife.45257.012

a dense amorphous layer and a filamentous layer (*Figure 1L–N*, *Figure 1—figure supplement 3F, G*) that may contain CotE, SpoIVA and other coat proteins that are recruited to the outer forespore membrane during engulfment (*McKenney et al., 2013*). Further studies are required to determine the molecular identity of these structures unambiguously.

## Septal PG is not completely degraded at the onset of engulfment

The cryo-FIB-ET images provided high-resolution details of the septum and the engulfing membrane. Hence, we focused on those details to obtain mechanistic insights about engulfment. The complete degradation of the septal PG

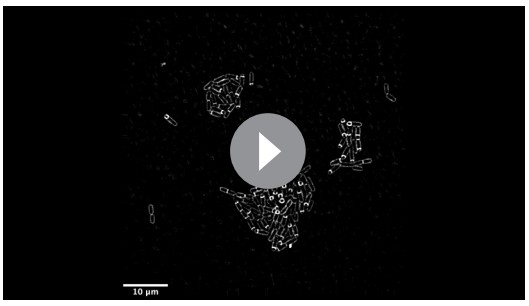

**Video 4.** Time-lapse microscopy of untreated sporulating *B. subtilis* cells stained with the membrane dye FM4-64. Pictures were taken every 5 minutes for 2 hours (related to *Figure 2—figure supplement 1*).

DOI: https://doi.org/10.7554/eLife.45257.013

during septal thinning has been traditionally considered a prerequisite for engulfment (*Abanes-De Mello et al., 2002*; *Chastanet and Losick, 2007*; *Eichenberger et al., 2001*; *Perez et al., 2000*). However, we observed a thin PG layer between the mother cell and the forespore membranes throughout engulfment in both wild type (*Figure 2A*; *Lopez-Garrido et al., 2018*) and a slender *ponA* mutant (*Tocheva et al., 2013*), suggesting that the septal PG is either not completely degraded or is quickly resynthesized after degradation.

To probe this, we imaged engulfing sporangia after treatment with antibiotics that block PG synthesis. When PG synthesis is inhibited, engulfment membrane migration does not continue, although the septum still stretches and curves into the mother cell (*Ojkic et al., 2016*). We reasoned that, if the septal PG was completely degraded, we would observe sporangia lacking PG between the mother cell and the forespore membranes after antibiotic treatment. However, if the septal PG was not completely degraded, the sporangia would show a layer of PG around the forespore, independent of antibiotic treatment.

To inhibit PG synthesis, we treated the cells with bacitracin, cephalexin, penicillin V or a combination of penicillin V and cephalexin (*Figure 2*). We previously assessed the extent of PG inhibition after antibiotic treatment by determining the frequency of division events, which rely on the synthesis of new PG (*Ojkic et al., 2016*). As expected, our results showed that untreated cells continued to grow and divide under the experimental conditions (*Figure 2—figure supplement 1A*) but when treated with cephalexin and penicillin V, cell division was completely blocked. These results also indicated that PG synthesis was inhibited within a few minutes of antibiotic treatment, because cells that were already undergoing septation were unable to complete septum formation (*Figure 2—figure supplement 1B–D*, top panels). To complement this analysis, we tested if these drugs also inhibited elongation, which also depends on PG synthesis (*Spratt, 1975*). To do so, we measured the elongation of vegetative cells present in sporulating cultures over a period of one hour after antibiotic treatment (*Figure 2—figure supplement 1E*, see Materials and methods). The length of wild type untreated vegetative cells increased by ~35% in an hour, but when treated with cephalexin or penicillin V alone, they elongated only ~10%. Furthermore, when treated with a combination of cephalexin and penicillin V, the cells elongated negligibly, indicating complete blockage of both septation and elongation (*Figure 2—figure supplement 1*, *Videos 4–7*).

For cryo-FIB-ET, we added the antibiotics two hours after inducing sporulation, when ~ 40–50% of the cells have undergone polar septation (*Ojkic et al., 2016*), and plunge froze the samples either one (for bacitracin) or two hours (for cephalexin, penicillin V, and combination of cephalexin and penicillin V) later. Indeed, we observed a thin PG layer in both untreated and antibiotic-treated sporangia (*Figure 2*, *Figure 2—figure supplement 2*), suggesting that septal PG is not completely degraded at the onset of engulfment.

## Septal thickness decreases slightly and uniformly across the entire septum during engulfment

The above observation prompted us to re-evaluate the process of septal thinning. The current model for septal thinning proposes that DMP initially localizes to the septal midpoint, where it starts degrading the septal PG as it moves towards the edge of the septal disk (*Abanes-De Mello et al., 2002*; *Chastanet and Losick, 2007*; *Illing and Errington, 1991*; *Meyer et al., 2010*). This enzymatic septal thinning model predicts that, during the transition from flat to curved septa, the septum should be thinner in the middle than at the edges (*Figure 3A*). To test this, we measured the distance between the forespore and the mother cell membranes across the length of the septum (see Materials and methods) for cells with flat (Stage II$_i$), curved (Stage II$_{ii}$) and engulfing (Stage II$_{iii}$) septa (*Figure 3B–D*, *Figure 3—figure supplement 1*). Sporangia with flat septa had an average septal thickness of ~23 nm±3.3 nm (*Figure 3B,E,F*) with 3 out of 5 septa being thicker at the middle (~28 nm) than at the edges (~22 nm) (*Figure 3B,E*, *Figure 3—figure supplement 1A*), contrary to what is proposed by the enzymatic septal thinning model. The septal thickness decreased by ~25% to~18 nm during later stages of engulfment (*Figure 3C–F*, *Figure 3—figure supplement 1B,C*) and a thin layer of PG was observed in all septa (*Figure 2A*, *Figure 2—figure supplement 2A–C*). Importantly, septal thickness was uniform across the entire septum during later stages (*Figure 3C,D,E*, *Figure 3—figure supplement 1B,C*) and no septum was thinner in the middle than in the edges. These results show that the transition from thick to thin septum is homogenous, contrary to the prediction of the

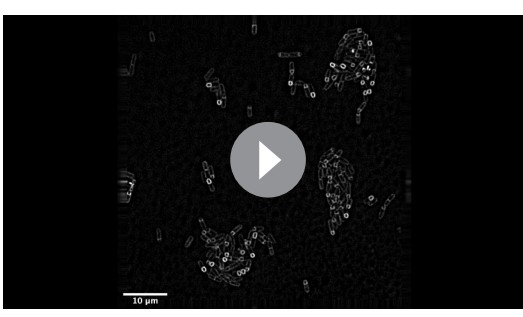

**Video 5.** Time-lapse microscopy of penicillin V-treated (500 µg/ml) sporulating *B. subtilis* cells stained with the membrane dye FM4-64. Pictures were taken every 5 minutes for 2 hours (related to *Figure 2—figure supplement 1*).
DOI: https://doi.org/10.7554/eLife.45257.014

enzymatic septal thinning model, and consistent with the model that DNA translocation dependent forespore growth stretches septal PG.

## SpoIIDMP is essential to maintain a thin, flexible septum

Next, we tested whether DMP was required to mediate the slight thinning observed during the transition from flat to curved septum. To address this question, we measured septal thickness in DMP mutants. In single mutants lacking D, M or P, engulfment is blocked but the septum bulges towards the mother cell, which complicates the analysis (*Figure 3—figure supplement 2*). However, bulge formation is largely abolished in a triple mutant lacking functional versions of D, M and P simultaneously (*Eichenberger et al., 2001*). Therefore, we imaged the DMP triple mutant by cryo-FIB-ET and measured its septal thickness (*Figure 3G–I*, *Figure 3—figure supplement 3*).

Most sporulation septa of the DMP triple mutant were either flat or slightly curved into the mother cell (*Figure 3G–I*, *Figure 3—figure supplement 3A–D*). In some cells, we observed membrane accumulation in the mother cell (*Figure 3H,I*, *Figure 3—figure supplement 3A,B*) and small bulges approximately in the middle of the septum (*Figure 3—figure supplement 3C,D*, *Video 8*). Septal thickness ranged from ~25 nm to ~45 nm, with an average thickness of 28 nm ±2.09 nm (*Figure 3E–G*, *Figure 3—figure supplement 3E*), which is ~5 nm greater than that of wild type sporangia with flat septa. Importantly, there were no significant differences in septal thickness between flat and curved septa in DMP mutant sporangia (*Figure 3G*, *Figure 3—figure supplement 3E*), indicating that DMP is in fact necessary for the slight thinning of the septum observed in wild type sporangia.

Surprisingly, the thickness of individual DMP mutant septum was irregular across the septal length, with thicker regions of more than 45 nm, that were not observed in wild type cells (*Figure 3G*, *Figure 3—figure supplement 3E*). One possible explanation for this finding is that in the absence of DMP, proteins involved in PG synthesis persist at the septum, and their continued activity leads to thicker and less flexible septal regions (*Figure 3J*). To test this model, we stained wild type and DMP mutant sporangia with BOCILLIN-FL, a fluorescent-derivative of penicillin V with affinity for multiple penicillin-binding proteins (PBPs) (*Zhao et al., 1999*, see Materials and methods). We observed continuous fluorescent signal around the septa in both strains, but the signal was

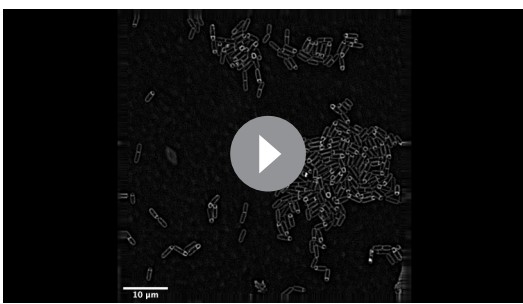

**Video 6.** Time-lapse microscopy of cephalexin-treated (50 µg/ml) sporulating *B. subtilis* cells stained with the membrane dye FM4-64. Pictures were taken every 5 minutes for 2 hours (related to *Figure 2—figure supplement 1*).
DOI: https://doi.org/10.7554/eLife.45257.015

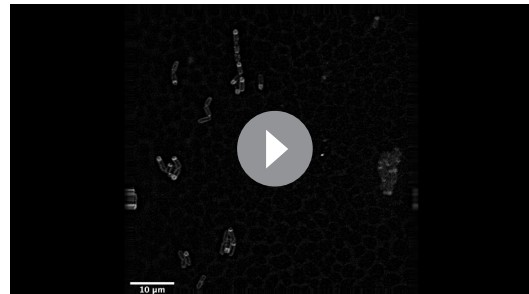

**Video 7.** Time-lapse microscopy of cephalexin- (50 µg/ml) and penicillin V- (500 µg/ml) treated sporulating *B. subtilis* cells stained with the membrane dye FM4-64. Pictures were taken every 5 minutes for 2 hours (related to *Figure 2—figure supplement 1*).
DOI: https://doi.org/10.7554/eLife.45257.016

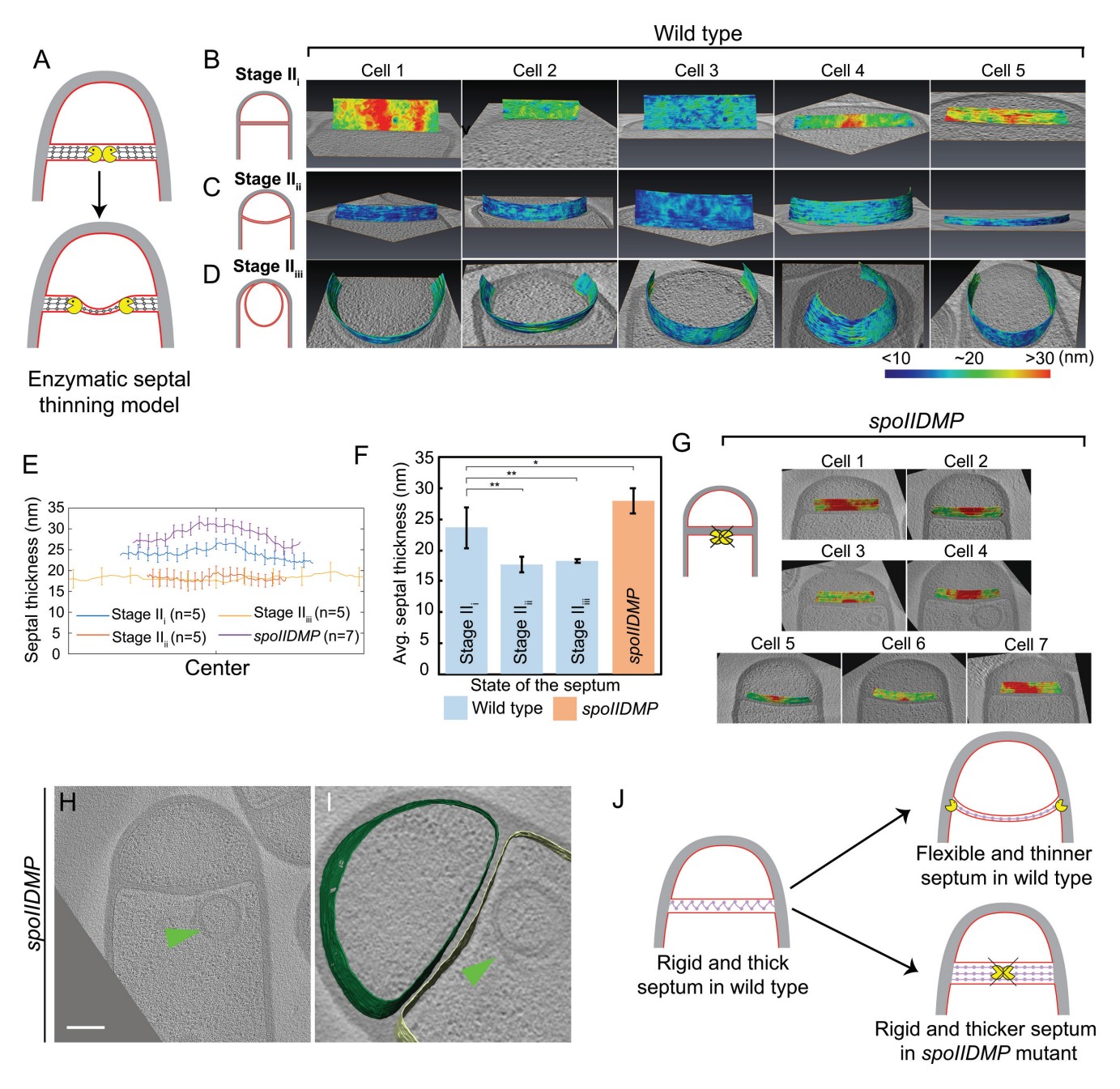

**Figure 3.** Septum is uniformly and only slightly thinned during engulfment and SpoIIDMP is required to maintain a thin flexible septum. (**A**) Schematic illustrating the model of septal thinning driven by septal PG (black meshwork) degradation by the DMP complex (yellow pacman) with membranes (red) and lateral PG (gray) highlighted. (**B–D**) Septal disc is color-coded according to the distance between the forespore and the mother cell membranes in five wild type sporangia with (**B**) flat, (**C**) curved and (**D**) engulfing septa. Schematic representing the morphology of each cell type is shown on the far left. (**E**) Average thickness of the septum across the forespore surface for the cells shown in (**B–D,G**). Error bars indicate standard deviation. n indicates the number of tomograms used for calculating septal distances in each case. (**F**) Average septal thickness for wild type flat (Stage II$_i$), curved (Stage II$_{ii}$) and engulfing (Stage II$_{iii}$) sporangia and *spoIIDMP* mutant sporangia. Error bars indicate standard deviation (n.s.: p>0.05; *: p≤0.05; **: p≤0.01; ***: p≤0.001, unpaired t-test). (**G**) Septal disc is color-coded according to the distance between the forespore and the mother cell membranes in seven *spoIIDMP* mutant sporangia. Scale bars have been omitted in distance plots as perspective views are shown. (**H**) Slice through a cryo-electron tomogram of *spoIIDMP* mutant sporangia. Scale bar: 200 nm. (**I**) Annotated forespore (green) and mother cell (yellow) membranes for the tomogram in (**H**). Excess membrane accumulation is highlighted by green arrows. (**J**) Schematic representing the role of DMP (yellow pacman) in septal thinning with membranes (red), lateral PG (gray) and septal PG (pink) highlighted. The rigid and thick septum in wild type can curve into the mother cell during engulfment but that of DMP becomes even thicker and does not curve into the mother cell.

*Figure 3 continued on next page*

*Figure 3 continued*

DOI: https://doi.org/10.7554/eLife.45257.017

The following source data and figure supplements are available for figure 3:

**Source data 1.** Raw data of septal distances in wild type *B. subtilis* sporangia and sporangia of engulfment-defective mutants used to plot the bar graph in *Figure 3F*.
DOI: https://doi.org/10.7554/eLife.45257.022

**Figure supplement 1.** Septal thickness during engulfment in wild type *B. subtilis*.
DOI: https://doi.org/10.7554/eLife.45257.018

**Figure supplement 2.** Septal thickness in *spoIIP* mutant sporangia.
DOI: https://doi.org/10.7554/eLife.45257.019

**Figure supplement 3.** Septal thickness in *spoIIDMP* sporangia.
DOI: https://doi.org/10.7554/eLife.45257.020

**Figure supplement 4.** Observing peptidoglycan synthesis using BOCILLIN-FL.
DOI: https://doi.org/10.7554/eLife.45257.021

brightest at the leading edge in wild type sporangia (*Ojkic et al., 2016*) and at different positions across the septum in DMP mutant sporangia (*Figure 3—figure supplement 4*). This mislocalization might allow ongoing synthesis of septal PG, leading to abnormally thick sporulation septa.

## PG is synthesized ahead of the leading edge of the engulfing membrane

Once the septum curves, the mother cell membrane starts to migrate around the forespore. Since the PG is not completely degraded, it will represent a major obstacle for the advancement of the engulfing membrane. To explain how cells overcome this hurdle, we previously proposed a conceptually new model for engulfment in which coordinated PG synthesis and degradation at the leading edge of the engulfing membrane moves the junction between the septum and the lateral cell wall around the forespore, making room for the engulfing mother cell membrane to advance (*Figure 1B*; *Ojkic et al., 2016*). In this 'make before break' model, new PG would be synthesized ahead of the engulfing membrane by forespore-associated PG biosynthetic complexes and subsequently degraded by DMP. To test this model, we first focused on the shape of the forespore membrane opposing the leading edge of the engulfing mother cell membrane. In wild type sporangia, the forespore membrane was rounded immediately ahead of the engulfing membrane (*Figure 4A*). This could be due to accumulation of additional PG at this site, which might push and deform the forespore membrane, introducing a broader curve. To confirm this, we analyzed the shape of the forespore membrane of sporangia in which PG synthesis was blocked with either bacitracin or cephalexin (*Figure 4B*, *Figure 4—figure supplement 1*, see Materials and methods). The forespore membrane was less rounded and had a sharp corner with a radius of curvature that was four times smaller than that of untreated cells (~30 nm vs. ~120 nm, *Figure 4C*). These results indicate that new PG is indeed synthesized ahead of the leading edge of the engulfing membrane, and that it deforms the forespore membrane at the junction between the septum and the lateral cell wall.

To exclude the possibility that new PG synthesis also happened behind the leading edge of the engulfing membrane, we analyzed septal thickness in sporangia treated with cephalexin and a combination of cephalexin and penicillin V using cryo-FIB-ET. If the septum is thinner in the presence of antibiotics, it would suggest that additional septal PG is synthesized after the DMP complex advances and degrades the PG ahead of the leading edge of the engulfing membrane. However, if septal thickness is independent of the presence of antibiotics, it would suggest that PG is not normally synthesized behind the DMP complex. Our data showed that the septal thickness of cells treated with cephalexin was comparable to that of untreated cells. Surprisingly, the septal thickness of cells treated with a combination of cephalexin and penicillin V was ~3 nm greater than that of untreated cells (*Figure 4—figure supplement 2*). This modest increase in septal thickness is of a magnitude that is consistent with recent molecular dynamics simulations which show that relaxed PG is thicker than stretched PG (*Beeby et al., 2013*). It is possible that treatment with antibiotics blocking PG synthesis inhibits the stretching of septal PG due to the absence of membrane migration which could pull the septal PG around the forespore. Alternatively, the residual DMP might partially cleave septal PG, releasing the tension and leading to slightly thicker septa. Taken together, in both cases of

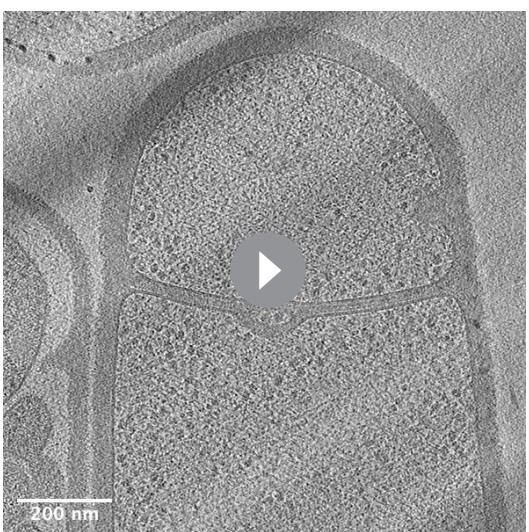

**Video 8.** Movie showing slices through a cryo-electron tomogram of *spoIIDMP* mutant sporangium shown in *Figure 3—figure supplement 3C*.
DOI: https://doi.org/10.7554/eLife.45257.023

antibiotic treatment the septal thickness did not decrease compared to untreated sporangia, suggesting that PG is not synthesized behind the DMP complex.

## 3D architecture of the leading edge of the engulfing membrane

The new PG synthesized ahead of the leading edge of the engulfing membrane might interfere with the movement of the engulfing membrane, until it is degraded by DMP. To obtain insights about the movement of the engulfing membrane, we annotated the forespore and the mother cell membranes in our tomograms to visualize the architecture of the leading edge in 3D. Our data showed that the leading edge of the engulfing membrane in wild type sporulating cells moved around the forespore in tiny finger-like projections (*Figure 4D–G*, *Figure 4—figure supplement 3*). The projections were ~10–30 nm wide and ~5–20 nm long, with significant variations from cell to cell. We hypothesized that those projections could be due to the uneven degradation of PG ahead of the leading edge of the engulfing membrane. To test this possibility, we imaged *spoIIP* mutant sporangia, which lack a functional DMP complex. As expected, the septum bulged towards the mother cell cytoplasm, but the mother cell membrane did not move forward in these cells. No membrane projections anywhere were observed in the mother cell membrane in any of the tomograms that were annotated (*Figure 4H–K*, *Figure 4—figure supplement 4*, *Video 9*). Also, no projections were observed in the DMP triple mutant, most of which did not form bulges (*Figure 4—figure supplement 5*). These findings suggest that SpoIIDMP is necessary for the formation of finger-like projections in the engulfing mother cell membrane.

We next tested whether PG synthesis was also required for the formation of finger-like projections. To study this possibility, we focused on sporangia in which PG synthesis was blocked using cephalexin (*Figure 4L–O*, *Video 10*). When treated with antibiotics that block PG synthesis, membrane migration is blocked although the forespore continues to grow into the mother cell (*Ojkic et al., 2016*). The case of cephalexin-treated cells is more complicated than other antibiotics, because after the septum curves into the mother cell, the leading edge sometimes retracts on one side while advancing slightly on the other (*Video 10*; *Ojkic et al., 2016*). This appears to consist of rotation of the 'cup' formed by the bulging septum relative to the lateral cell wall, rather than membrane migration, because the distance between the leading edges does not decrease during this process. Thus, rotation of the septal cup does not reflect the degree to which the forespore is engulfed. Cephalexin inhibits the earliest stages of cell division (*Eberhardt et al., 2003*; *Kocaoglu and Carlson, 2015*), and therefore we speculate that it might be required to tether the extending septum to the lateral cell wall. In the absence of these bridges, the septum might be free to rotate according to Brownian motion, perhaps anchored by the Q-AH ratchet that can also mediate engulfment in the absence of the cell wall (*Broder and Pogliano, 2006*). We used cryo-FIB-ET to compare the architecture of both sides of the engulfing membrane in cephalexin-treated sporangia (*Figure 4L–O*). We did not observe finger-like projections in the side of the membrane that retracts (*Figure 4M*) but observed a few projections in the opposite side (*Figure 4N*), which might remain tethered to PG ahead of the engulfing membrane. Also, when cells were treated with penicillin V, we observed fewer projections that were shorter compared to untreated cells (*Figure 4—figure supplement 6*). Taken together, these results suggest that the finger-like projections at the leading edge of the engulfing membrane might be caused by tethering of the engulfing membrane to the PG via DMP (*Figure 4P*).

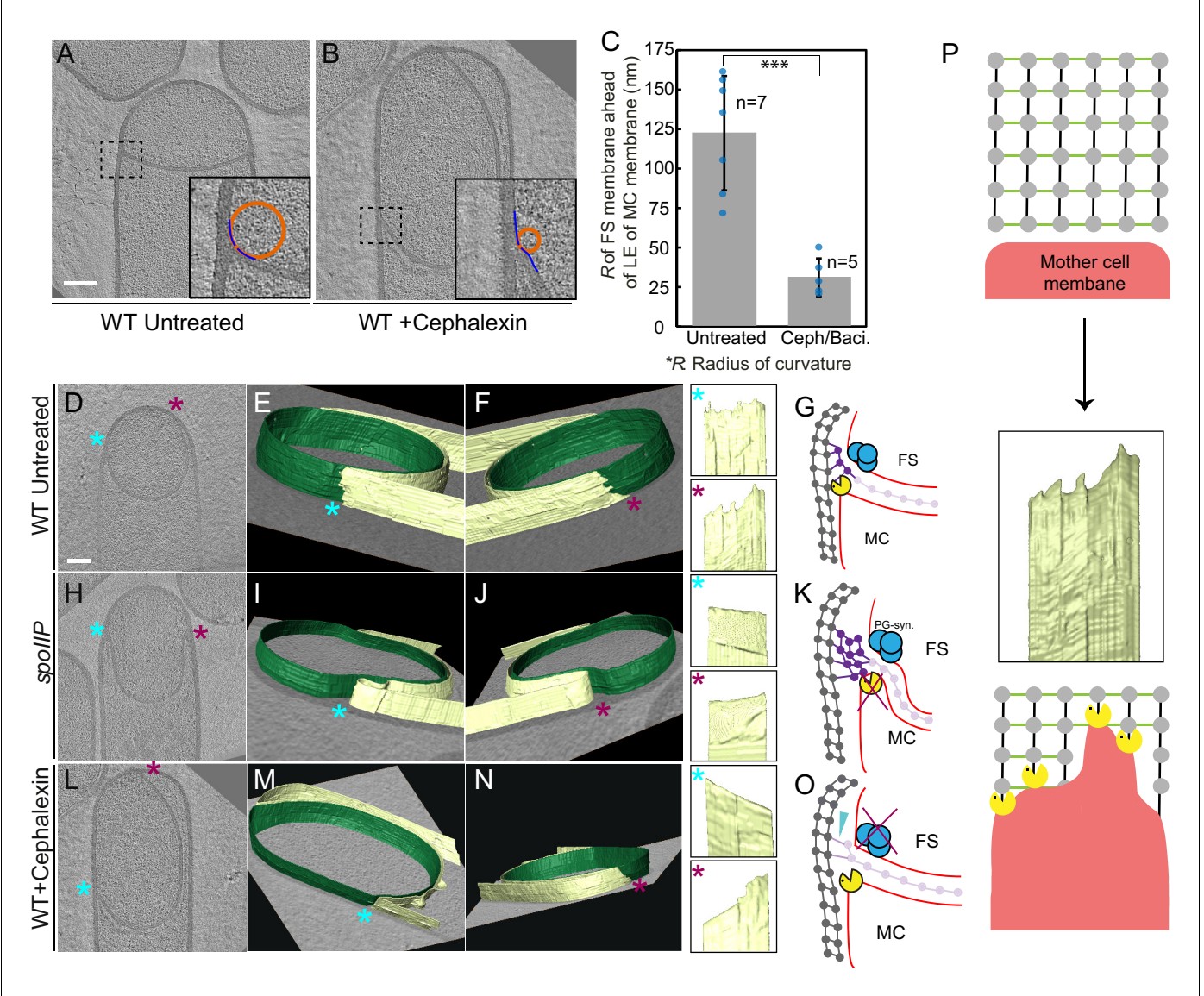

**Figure 4.** Architecture of the leading edge of the engulfing membrane. (A,B) Slices through cryo-electron tomograms of wild type (A) untreated and (B) cephalexin-treated sporangia. The radius of curvature (orange circle) of the forespore membrane (blue spline) ahead of the leading edge of the engulfing membrane is highlighted in the respective insets. (C) Plot showing the average radius of curvature (nm) of the forespore membrane ahead of the leading edge of the engulfing membrane for untreated and antibiotic-treated sporangia. Blue dots indicate individual data points (as also indicated by n); black bars indicate the standard deviation (***: p<0.001, unpaired t-test). (D) Slice through a cryo-electron tomogram of wild type *B. subtilis* sporangium. (E,F) Annotated forespore (green) and mother cell (yellow) membranes for the tomogram shown in (D) as viewed from both the left (blue asterisk) and the right (maroon asterisk) sides respectively, with insets of zoomed-in views of the leading edge of the engulfing membrane of both sides. Similar labeling scheme is followed through *(H–N)*. (G) Schematic showing the localization of DMP PG degradation machinery (yellow pacman) and PG synthases (blue circles). Membranes (red), lateral PG (gray), septal PG (pink) and new PG (purple) are also highlighted. (H) Slice through a cryo-electron tomogram of *spoIIP* mutant sporangium. (I,J) Annotated membranes for the tomogram shown in (H) with insets of zoomed-in views of the leading edge of the engulfing membrane of both sides. (K) Schematic representing a cell in which the DMP complex (yellow pacman) does not assemble. (L) Slice through a cryo-electron tomogram of cephalexin-treated sporangium. (M,N) Annotated membranes for the tomogram shown in (L) with insets of zoomed-in views of the leading edge of the engulfing membrane of both sides. (O) Schematic representing a cell in which PG synthesis (blue circles) has been inhibited. Scale bar for (D,H,L): 200 nm. Scale bars have been omitted for surface rendered images owing to their perspective nature. (P) Model for mother cell membrane migration. DMP complexes (yellow pacman) present at different positions on the mother cell membrane (red) tether the membrane to the PG (gray) synthesized ahead. Due to a limited number of DMP complexes, the engulfing membrane may move forward in finger-like projections. This is indicated by a representative annotated mother cell membrane (yellow) from *Figure 4F*.

DOI: https://doi.org/10.7554/eLife.45257.024

The following figure supplements are available for figure 4:

*Figure 4 continued on next page*

*Figure 4 continued*

**Figure supplement 1.** Quantification of radius of curvature of the forespore membrane.

DOI: https://doi.org/10.7554/eLife.45257.025

**Figure supplement 2.** Graph depicting the septal thickness of wild type untreated and antibiotic-treated sporangia (cephalexin and a combination of cephalexin and penicillin V) according to concentrations specified in *Figure 2—figure supplement 1*.

DOI: https://doi.org/10.7554/eLife.45257.026

**Figure supplement 3.** Membrane architecture in wild type.

DOI: https://doi.org/10.7554/eLife.45257.027

**Figure supplement 4.** Membrane architecture in spoIIP sporangia.

DOI: https://doi.org/10.7554/eLife.45257.028

**Figure supplement 5.** Membrane architecture in *spoIIDMP* sporangia.

DOI: https://doi.org/10.7554/eLife.45257.029

**Figure supplement 6.** Membrane architecture in antibiotic-treated sporangium.

DOI: https://doi.org/10.7554/eLife.45257.030

## Discussion

In this study, we have visualized the developmental process of sporulation in *Bacillus subtilis* using cryo-FIB-ET. We have obtained images of cells during different stages of sporulation at a resolution of a few nanometers, revealing new details about the architecture of spore assembly, as well as several hitherto unknown structures inside and around the developing spore (*Figure 1*). Our results also provide mechanistic insights into engulfment, including the early step of septal thinning (*Figures 2* and *3*) and membrane migration (*Figure 4*) which are captured in the model presented in *Figure 5*.

We provide evidence that the septal PG is not completely degraded at the onset of engulfment. Instead, the septum gets slightly (~25%) thinner as it curves into the mother cell, with PG continuously present between the mother cell and the forespore membranes. In addition, the whole septum transitions from thick to thin homogenously, contrary to previous results that suggested that septal thinning started in the middle and progressed towards the edges. The homogenous thinning of the septum can be explained by our previous finding that septal PG is stretched as the forespore grows towards the mother cell due to increased turgor pressure caused by SpoIIIE-mediated chromosome translocation (*Lopez-Garrido et al., 2018*). Thus, septal thinning could simply represent a change in the conformation of the septal PG, from a relaxed to a stretched state, triggered by the increased turgor pressure in the forespore. This is consistent with recent molecular dynamics simulations on Gram-positive cell walls, which show that relaxed PG fragments are ~25% thicker than those in a strained conformation (*Beeby et al., 2013*).

Our results also indicate that DMP is required to maintain a flexible septum that can curve as the forespore grows into the mother cell (*Figure 3*). We found that the septa of DMP triple mutants have irregular thickness and are on average thicker than wild type septa (*Figure 3E–G*). Since DMP is produced after polar septation, the septa of DMP mutant sporangia must have thickened after they have been formed. Thus, it is possible that DMP prevents septal thickening by clearing PG synthases from the septum, where they would accumulate after polar septation. This model is consistent with the increased

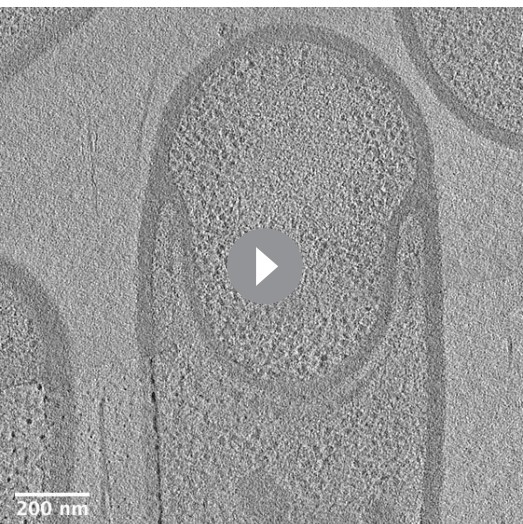

**Video 9.** Movie showing slices through a cryo-electron tomogram of *spoIIP* mutant sporangium shown in *Figure 4H*.

DOI: https://doi.org/10.7554/eLife.45257.031

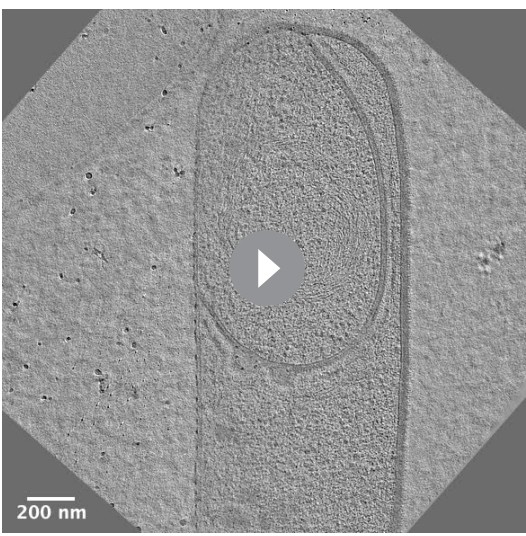

200 nm

**Video 10.** Movie showing slices through a cryo-electron tomogram of wildtype *B. subtilis* sporangium treated with cephalexin shown in *Figure 4L*.
DOI: https://doi.org/10.7554/eLife.45257.032

localization of PBPs throughout the septum in DMP mutants (*Figure 3—figure supplement 4*). Nevertheless, we cannot rule out the possibility that DMP degrades the septal PG partially to generate a flexible septum. If this was the case, the partial degradation should happen simultaneously across the whole septum to enable the homogenous transition from thick to thin. However, since DMP is rate limiting for engulfment, it seems unlikely that there may be enough DMP complexes to mediate the homogeneous thinning of the septum. We therefore favor the model that septal thinning is primarily driven by stretching of septal peptidoglycan.

Our data also provide insights into the mechanism of membrane migration during engulfment. Using fluorescence microscopy, we previously observed that new PG is synthesized at the leading edge of the engulfing membrane (*Ojkic et al., 2016*; *Tocheva et al., 2013*). We also showed that many forespore penicillin binding proteins (PBPs) can track the leading edge of the engulfing mother cell membrane (*Ojkic et al., 2016*), suggesting that PG synthesis at the leading edge of the engulfing membrane is carried out by forespore PBPs. The cryo-FIB-ET images presented here provide further support to this idea: the comparison of the architecture of the leading edge of the engulfing membrane between native sporangia and sporangia in which PG synthesis is blocked by antibiotics shows that new PG deforms and rounds the forespore membrane ahead of the leading edge of the engulfing membrane (*Figure 4A–C*), indicating that new PG is synthesized by forespore PBPs immediately ahead of the leading edge of the engulfing mother cell membrane. We propose that DMP complexes at the leading edge of the engulfing membrane target this new PG for degradation, making room for the engulfing membrane to advance (*Figures 1B* and *5C*). This relatively simple model suggests that engulfment could have evolved by developing new mechanisms to spatially and temporally regulate the conserved protein machineries that synthesize and degrade peptidoglycan.

The 3D reconstruction of the leading edge of the engulfing mother cell membrane shows the presence of finger-like projections that resemble the filopodia of eukaryotic cells (*Mattila and Lappalainen, 2008*). In eukaryotic cells, these membrane projections are produced by cytoskeletal proteins, and the projections are typically a few micrometers wide. In contrast, no cytoskeletal elements contributing to engulfment have been described so far (or visualized in our tomograms) and the finger-like projections at the leading edge of the engulfing membrane are only a few nanometers wide. Instead, our results demonstrate that DMP is required for the formation of these finger-like projections. The simplest model to explain these data is that DMP tethers the engulfing membrane to the PG ahead of the leading edge of the engulfing membrane, as it degrades the new PG to make room for the engulfing membrane to expand. DMP activity is rate limiting for membrane migration (*Abanes-De Mello et al., 2002*; *Meyer et al., 2010*), suggesting that there is a discrete number of DMP complexes to remove the steric barrier posed by the newly synthesized PG ahead of the leading edge. The limited number of DMP complexes might cause the PG to be degraded irregularly, generating an uneven membrane front in the form of finger-like projections. The distance between the tips of the finger-like projections is ~5–20 nm, which would require that some DMP complexes are ahead of others by 2 to 8 glycan strands (*Turner et al., 2018*). Our observations are consistent with the hypothesis that the finger-like projections are produced via the tethering of the DMP complex to existing peptidoglycan, since short fingers are observed when PG synthesis is inhibited, and that these fingers are either stabilized or enhanced by binding newly synthesized peptidoglycan, since longer fingers are observed in the presence of ongoing PG synthesis. This is supported by prior studies of the two enzymes that degrade peptidoglycan (D and P), which have

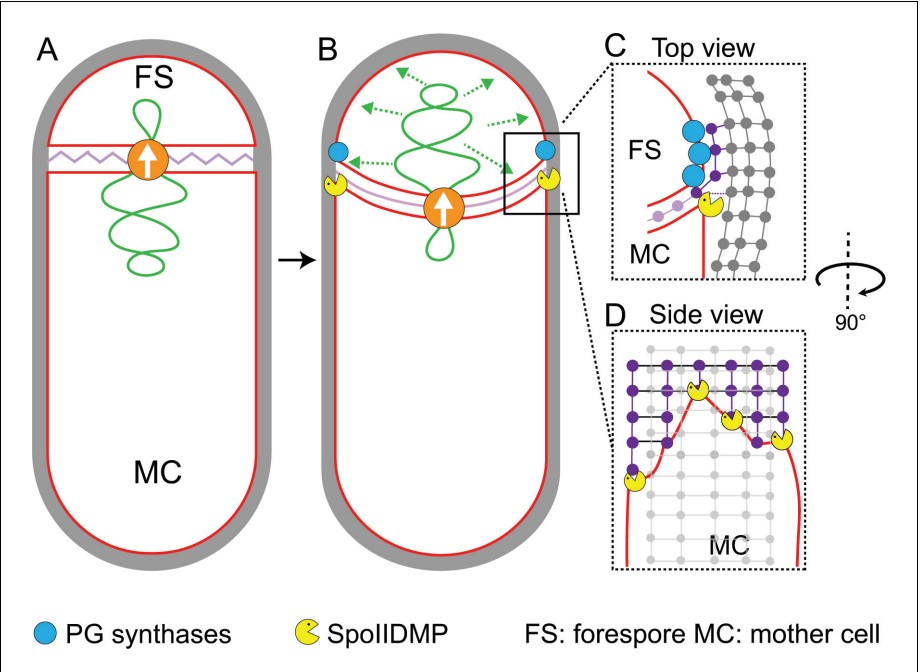

**Figure 5.** Model for septal thinning and membrane migration. (**A**) Schematic of a sporulating cell with a flat septum and relaxed septal PG (pink). Membranes (red), lateral PG (gray), SpoIIIE (orange) and forespore chromosome (green) are also highlighted. (**B**) As SpoIIIE translocates the chromosome to the forespore, the septal PG is stretched which may contribute to septal thinning. (**C**) Top view of the sporulating cell wherein coordinated PG degradation by DMP (yellow pacman) and PG synthesis by PG biosynthetic enzymes (blue) at the leading edge provide room for the engulfing mother cell membrane migration. (**D**) Side view depicting the proposed membrane migration model with mother cell membrane (red), PG synthesized ahead of the membrane (purple) and lateral PG (gray) highlighted. During engulfment, DMP complexes present at different locations on the mother cell membrane may move forward by degrading the PG ahead and their rate-limiting activity may lead to the formation of tiny finger-like projections. FS: Forespore; MC: mother cell.

DOI: https://doi.org/10.7554/eLife.45257.033

shown that these comprise a processive enzyme complex in which P binds and cleaves PG first, allowing D to bind and cleave PG (*Morlot et al., 2010*). Our cell biological data showed that localization of P to the leading edge of the engulfing membrane is decreased but not completely absent when PG synthesis is inhibited (*Ojkic et al., 2016*), suggesting that P can bind to existing PG, but that high affinity binding requires newly synthesized peptidoglycan. This increased binding of P would likely allow the formation of longer finger-like projections. From a functional perspective, we speculate that the finger-like projections could be compared to 'friction ridges', the minutely raised ridges of the epidermis that provide a grasping surface on our fingers (*Tomlinson et al., 2007*). During engulfment, these membrane projections may provide the engulfing membrane a tighter lateral grip while moving around the forespore, hence serving as a ratchet that prevents backward movement of the membrane.

The application of cryo-FIB-ET has been instrumental in allowing us to visualize and accurately measure structural details of engulfing sporangia, which transpire at a scale of a few nanometers. Our studies support a mechanistic model for the enigmatic process of engulfment, while revealing novel architectural details about engulfment and spore assembly, including intriguing new molecular structures that will require further study to unambiguously identify them. Our findings reveal details about sporulation at an unprecedented resolution and further illustrate the potential of cryo-FIB-ET to reveal critical new information about dynamic biological processes.

# Materials and methods

## Key resources table

| Reagent type (species) or resource | Designation | Source or reference | Identifiers | Additional information |
|---|---|---|---|---|
| Strain, strain background (*Bacillus subtilis* PY79) | PY79 | (*Youngman et al., 1984*) | Tax. ID:1415167 | Wild type |
| Strain, strain background (*Bacillus subtilis* PY79) | Δ*spoIIP::tet* | (*Frandsen and Stragier, 1995*) | | KP513 |
| Strain, strain background (*Bacillus subtilis* PY79) | *spoIID298, spoIIM-mls, ΔspoIIP::tet* | (*Broder and Pogliano, 2006*) | | KP4188 |
| Chemical compound, drug | FM4-64 | Thermo Fisher Scientific | Cat#T13320 | |
| Chemical compound, drug | Bacitracin | MilliporeSigma | Cat#B0125 | Conc. used: 50 µg/ml |
| Chemical compound, drug | Penicillin V | MilliporeSigma | Cat#1504489 | Conc. used: 500 µg/ml |
| Chemical compound, drug | Cephalexin | MilliporeSigma | Cat#C4895 | Conc. used: 50 µg/ml |
| Software, algorithm | JFilament | (*Smith et al., 2010*) | http://athena.physics.lehigh.edu/jfilament/ | |
| Software, algorithm | IMOD | (*Mastronarde, 1997*) | http://bio3d.colorado.edu/imod/; RRID: SCR_003297 | |
| Software, algorithm | TomoSegMemTV | (*Martinez-Sanchez et al., 2014*) | https://sites.google.com/site/3demimageprocessing/tomosegmemtv | |
| Software, algorithm | Amira | Commercial software by Thermo Scientific (formerly FEI) | https://www.fei.com/software/amira-3d-for-life-sciences/; RRID: SCR_014305 | |
| Software, algorithm | SerialEM | (*Mastronarde, 2005*) | http://bio3d.colorado.edu/SerialEM/ | |
| Software, algorithm | Matlab code to calculate septal distances of sporangia | This paper; *Source code 1* | | |
| Software, algorithm | Matlab code to calculate cell length using fluorescence microscopy images | This paper; *Source code 2* | | |

## Strains and culture conditions

We used *Bacillus subtilis* PY79 background for all data acquisition. The strains (see *Key Resources Table*) were grown in LB plates at 30°C. The bacteria were first grown in ¼ diluted LB to OD$_{600}$ ~0.5–0.7. Sporulation was then induced by resuspension in A + B media at 37°C. For cryo-FIB-ET, we collected wild type sporulating cells 1.5–3 hr after sporulation induction. For cells treated with antibiotics, 50 µg/ml of bacitracin, 50 µg/ml of cephalexin and 500 µg/ml of penicillin V were used. Antibiotics were added two hours after induction of sporulation and samples for cryo-FIB-ET were collected either one (for bacitracin) or two hours (for cephalexin, penicillin V and a combination

of cephalexin and penicillin V) later. For *spoIIP* and *spoIIDMP* mutant sporangia, cells were collected 2.5 hr after induction of sporulation for cryo-ET.

## Cryo-FIB-ET and image processing

7 µl of diluted liquid culture was deposited onto glow-discharged (using Pelco easyGlow) holey carbon coated QUANTIFOIL R 2/1 200 mesh copper grids. Manual blotting was performed using Whatman No. one filter paper from the reverse side to remove excess resuspension media such that cells form a monolayer on the grids. They were then immediately plunge-frozen into a liquid ethane/propane mixture cooled by liquid nitrogen using a custom-built vitrification device (Max Planck Institute for Biochemistry, Munich). These grids were then stored in storage boxes in liquid nitrogen until further use.

Vitrified bacterial samples forming a monolayer on the grids were mounted into modified autogrids (Max Planck Institute for Biochemistry) for milling inside a Thermo Scientific Scios DualBeam (cryo-FIB/SEM) (Materials and methods). 100–250 nm thin sections or lamellae (12–15 µm in width) were then prepared using rectangular milling patterns and beam current of 0.1 nA or 0.3 nA for rough milling and 10 pA or 30 pA for fine milling as described in *Chaikeeratisak et al. (2017)*. Tilt series were collected in a 300-keV Tecnai G2 Polara (Thermo Scientific) equipped with a K2 Summit direct detection camera (Gatan) and a post-column Quantum energy filter (Gatan). The samples were usually tilted from −66˚ to +66˚ (the range per tilt series depending on the quality of the lamellae) with an increment of 1.5˚ or 2˚. The tilt series were acquired using SerialEM (*Mastronarde, 2005*). The images were recorded at a defocus of −5 to −6 µm at nominal magnifications of 34,000 (pixel size: 0.61 nm) or 41,000 (pixel size: 0.49 nm) with a cumulative dose of ~60–130 e⁻/Å². Images for later stages of sporulation (*Figure 1—figure supplement 2B*) were acquired at nominal magnification of 22,500 (pixel size: 0.92 nm) as these samples were highly sensitive to radiation damage.

The patch-tracking feature of IMOD was used to reconstruct the tilt-series (*Kremer et al., 1996*). TomosegmemTV (*Martinez-Sanchez et al., 2014*) was used for semi-automatic segmentation of membranes followed by manual refinement in the Amira software package.

## Measuring mean radius of ellipsoidal complexes

The mean radius (r) of an ellipse is given by:

$$r = \sqrt{\frac{a^2 + b^2}{2}}$$

where *a* is the semi-major axis of the ellipse and *b* is the semi-minor axis of the ellipse. For ellipsoidal complexes observed in our tomograms (*Figure 1J*, *Figure 1—figure supplement 3A–C*), average value of *a* is ~45 nm and that of *b* is ~42 nm.

## Calculating septal distances

'Surface distance' feature of Amira was used to calculate the septal distances between the forespore and mother cell membranes. The septal-disc was color coded according to these values (*Figure 3B–D,G*, *Figure 3—figure supplement 2E*). Five tomograms each of wild type cells representing different stages of engulfment (flat, curved and engulfing septa) were used for analysis. For engulfment mutants, five *spoIIP* sporangia and seven *spoIIDMP* sporangia were analyzed. For antibiotic treated cells, six cephalexin-treated sporangia and five cephalexin- and penicillin V- combination treated sporangia were analyzed. To get the linear profiles of the distances, the data was grouped into smaller bins of approximately equal area for each tomogram. Then the average of the distance within a bin was used to represent the distance between the membranes at that location of the bin (see *Source code 1*). *Figure 3E* represents an average of all the profiles for each cell type in *Figure 3—figure supplement 1* and *Figure 3—figure supplement 3E*.

## Fluorescence microscopy

Cells were visualized on an Applied Precision DV Elite optical sectioning microscope equipped with a Photometrics CoolSNAP-HQ2 camera and deconvolved using SoftWoRx v5.5.1 (Applied Precision). For experiment outlined in *Figure 3—figure supplement 4*, the median focal plane of the image is shown. Membranes were stained with 0.5 µg/ml of FM4-64 (Thermo Fisher Scientific) that was added

directly to 1.2% agarose pads prepared using sporulation resuspension medium. 10 µg/µl of BOCIL-LIN-FL was added to 1 ml of culture aliquoted ~2.5 hr after sporulation induction (at 37˚C) and washed with sporulation resuspension medium three times. 12 µl of washed cells were then transferred to agarose pads for imaging.

For time-lapse microscopy, sporulation was induced at 30˚C. 0.5 µg/ml FM4-64 was added to the cultures ~1.5 hr after sporulation induction and incubation continued for another hour. Composition of agarose pads for time-lapse microscopy is as follows: 2/3 vol of supernatant from the sporulation culture, 1/3 vol 3.6% agarose in fresh A + B sporulation medium, 0.17 µg/ml FM4-64, supplemented with antibiotics according to concentrations mentioned above in 'Strains and culture conditions'. 12 µl samples were taken 3 hr after resuspension and transferred to the agarose pads. Pads were covered with a glass slide and sealed with petroleum jelly to avoid dehydration during time-lapse imaging. Pictures were taken in an environmental chamber at 30˚C every 5 min for ~5 hr. Excitation/emission filters were TRITC/CY5 for membrane imaging. Excitation light transmission was set to 5% to minimize phototoxicity and exposure time was set to 0.1 s.

### Calculating cell length using fluorescence microscopy images

To determine the length of vegetative cells over time (Figure 2—figure supplement 1E), membrane contours were extracted from microscopy images for each time frame (up to 60 min post treatment with antibiotics) using semi-automated active contour software JFilament, available as a Fiji plugin (Schindelin et al., 2012; Smith et al., 2010). The cell length was then calculated by measuring along the long axis of the contours using a custom-built MATLAB script (see Source code 2). To plot the average cell length, the data was normalized to the initial cell length for each of the cases.

### Calculating radius of curvature

To calculate radius of curvature (Figure 4A–C, Figure 4—figure supplement 1), a slice was taken approximately from the center of the z-stack for each of the tomograms. 'Measure spline curvature' feature of sabl_mpl (Yao et al., 2017) was then used to plot radii of curvatures around the forespore membranes just ahead of the leading edge for seven native sporangia and five antibiotic-treated sporangia wherein membrane migration appears to be blocked.

## Acknowledgements

This work was supported by National Institutes of Health Director's New Innovator Award 1DP2GM123494 (EV), and the National Institutes of Health R01-GM057045 (EV and KP). We used the UC San Diego Cryo-EM Facility (partially supported by a gift from the Agouron Institute to Tim Baker), and the San Diego Nanotechnology Infrastructure of UC San Diego (supported by the NSF grant ECCS-1542148). The authors would like to thank Antonio Martínez-Sánchez for useful insights into data analysis.

## Additional information

### Funding

| Funder | Grant reference number | Author |
| --- | --- | --- |
| National Institutes of Health | National Institute of Health's Director's New Innovator Award (1DP2GM123494) | Elizabeth Villa |
| National Institutes of Health | RO1-GM057045 | Kit Pogliano Elizabeth Villa |

The funders had no role in study design, data collection and interpretation, or the decision to submit the work for publication.

## Author contributions
Kanika Khanna, Javier Lopez-Garrido, Conception and design, Data acquisition, Data analysis, Writing—original draft; Ziyi Zhao, Yuan Yuan, Joseph Sugie, Data analysis; Reika Watanabe, Assistance with cryo-EM; Kit Pogliano, Elizabeth Villa, Conception and design, Data analysis, Writing—review and editing

## Author ORCIDs
Kanika Khanna (iD) https://orcid.org/0000-0001-7150-0350
Javier Lopez-Garrido (iD) https://orcid.org/0000-0002-8907-502X
Ziyi Zhao (iD) https://orcid.org/0000-0003-4455-3224
Reika Watanabe (iD) http://orcid.org/0000-0002-7427-7702
Yuan Yuan (iD) https://orcid.org/0000-0002-9509-9167
Joseph Sugie (iD) https://orcid.org/0000-0003-2911-1807
Kit Pogliano (iD) https://orcid.org/0000-0002-7868-3345
Elizabeth Villa (iD) https://orcid.org/0000-0003-4677-9809

## Decision letter and Author response
Decision letter https://doi.org/10.7554/eLife.45257.044
Author response https://doi.org/10.7554/eLife.45257.045

# Additional files

## Supplementary files
• Source code 1. Matlab code to calculate septal distances of sporangia.
DOI: https://doi.org/10.7554/eLife.45257.034
• Source code 2. Matlab code to calculate cell length using fluorescence microscopy images.
DOI: https://doi.org/10.7554/eLife.45257.035
• Transparent reporting form
DOI: https://doi.org/10.7554/eLife.45257.036

## Data availability
The authors have created a library of *B. subtilis* tomograms accessible at: http://villalab.ucsd.edu/research/engulfment. The authors have also deposited three representative tilt-series to Electron Microscopy Data Bank (EMDB) in the form of 4x binned tomograms. The IDs are EMD-20335, EMD-20336, EMD-20337 for Figure 1D,F,H respectively.

The following datasets were generated:

| Author(s) | Year | Dataset title | Dataset URL | Database and Identifier |
|---|---|---|---|---|
| Khanna K, Lopez-Garrido J, Zhao Z, Watanabe R, Yuan Y, Sugie J, Pogliano K, Villa E | 2019 | Representative tilt-series | http://www.ebi.ac.uk/pdbe/entry/emdb/EMD-20335 | Electron Microscopy Data Bank, EMD-20335 |
| Khanna K, Lopez-Garrido J, Zhao Z, Watanabe R, Yuan Y, Sugie J, Pogliano K, Villa E | 2019 | Representative tilt-series | http://www.ebi.ac.uk/pdbe/entry/emdb/EMD-20336 | Electron Microscopy Data Bank, EMD-20336 |
| Khanna K, Lopez-Garrido J, Zhao Z, Watanabe R, Yuan Y, Sugie J, Pogliano K, Villa E | 2019 | Representative tilt-series | http://www.ebi.ac.uk/pdbe/entry/emdb/EMD-20337 | Electron Microscopy Data Bank, EMD-20337 |

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
