## [Decision Letter]

Thank you for sending your article entitled "The molecular architecture of engulfment during *Bacillus subtilis* sporulation" for peer review at *eLife*. Your article is being evaluated by two peer reviewers, and the evaluation is being overseen by a Reviewing Editor and Gisela Storz as the Senior Editor.

Engulfment is a phagocytic-like process that is essential for bacterial sporulation. During engulfment, the mother cell membrane migrates around the developing spore, generating a cell within a cell. Membrane migration involves cell wall synthesis and remodeling events that remain incompletely resolved. In this study, authors use state-of-the-art cryo-FIB-ET to investigate these unresolved aspects in *Bacillus subtilis*. Their data confirm earlier observations (obtained by ECT, Tocheva et al., 2013) that a layer of PG remains in the intermembrane space separating the mother cell and forespore. In addition, the unprecedented resolution of their images provides new insights into the thickness of this septal PG layer, as well as the morphology of the engulfing membranes at different stages of the engulfment process and upon impaired PG degradation or synthesis.

Essential revisions:

While their observations represent a significant contribution to the field, there are two major concerns regarding their conclusions, both based on the fact that the extend of PG inhibition in the presence of antibiotics is not clear:

- Authors observe finger-like projections associated with migrating membranes. They provide convincing evidence that these projections are associated with the DMP complex and they propose that they result from "tethering of the engulfing membrane to newly synthesized PG via DMP". However, these finger-like projections are still observed at the leading side of the engulfing membrane, although being less prominent, when PG synthesis is inhibited by cephalexin (Figure 4N and subsection “3D architecture of the leading edge of the engulfing membrane”, second paragraph).

Do authors really consider that PG synthesis is fully inhibited in the presence of cephalexin? If so, remnant finger-like projections and membrane migration still occurs when PG synthesis is inhibited. They might result from tethering of the engulfing membrane to old PG surrounding the forespore via DMP. In that case, finger-like projections may be stabilized/enhanced (rather than "caused") by tethering of the engulfing membrane to newly synthesized PG. To resolve this problem, the authors need to test the extent of PG synthesis inhibition by Cephalexin.

- Figure 4B shows a reduced rounding of the forespore membrane in the presence of antibiotics, providing evidence that PG is synthesized at the leading edge of the engulfing membrane. Based on this observation and a previous study (Ojkic et al., 2016), authors propose a model (Figure 5) in which PG is exclusively synthesized ahead of the leading edge of the engulfing membrane. However, these data do not exclude the possibility that PG synthesis also happens behind the leading edge of the engulfing membrane, or in other words that PG is resynthesized after degradation by the DMP complex. Indeed, as pointed in the previous point, the fact that membrane migration still happens in the presence of antibiotics again questions whether PG synthesis is fully inhibited. In cells displaying incomplete or asymmetrical engulfment (such as those shown in Figure 2), PG synthesis might have continued for a while and stopped, resulting in septal disks containing newly synthesized PG and leading edges showing impaired PG synthesis. Analysis of septum thickness (as performed in Figure 3B-E) in the presence of antibiotics might shed light on this issue: a thinner septum (but not absent if DMP does not totally degrades septal PG) in the presence of antibiotics would suggest that PG is synthesized behind the DMP complex. If septum thickness is independent of the presence of antibiotics, it suggests that either PG is not synthesized behind the DMP complex or that PG is not fully inhibited in the presence of antibiotics. This uncertainty impacts the hypothesis proposed by the authors because PG synthesis happening behind the DMP complex might explain why a slight and homogenous septal thinning is observed (Figure 3); it might also contribute to the residual PG layer observed in the intermembrane space

---

## [Author Response]

This revised version includes additional experiments and analysis to address the reviewers’ comments. Here is a summary of the main points: We did additional experiments to more precisely test the extent of peptidoglycan (PG) synthesis inhibition upon antibiotic treatment to ensure that our results are representative. In addition to using septation as a readout for PG synthesis, we determined the rate of elongation of vegetative cells when treated with different antibiotics and tested two additional drug treatments – penicillin V and a combination of cephalexin and penicillin V. We found that a combination of cephalexin and penicillin V was most effective to ensure complete block of both septation and elongation (Figure 2—figure supplement 1, Videos 7-10). We also acquired cryo-electron tomography data of wild type sporangia treated with penicillin V and a combination of cephalexin and penicillin V (Figure 2, Figure 2—figure supplement 2), measured the septal thickness of sporangia treated with cephalexin and a combination of cephalexin and penicillin V, and compared it to untreated sporangia (Figure 4—figure supplement 2). The data showed that the septal thickness did not decrease upon antibiotic treatment which suggested that PG synthesis did not precede the mother cell DMP complex. In addition, we have modified the text wherever necessary to explain the impact of antibiotic treatment on septal architecture and engulfment in a more detailed and clear manner.

Essential revisions:While their observations represent a significant contribution to the field, there are two major concerns regarding their conclusions, both based on the fact that the extend of PG inhibition in the presence of antibiotics is not clear:

We agree that determining the extent of PG inhibition in the presence of antibiotics is key for the interpretation of our results. We explain below the experiments we have already done to determine this, and propose additional experiments to more precisely test the extent of PG synthesis inhibition. We will also collect additional cryo-electron tomograms of sporulating cells treated with additional PG inhibitors to ensure that our results are representative.

We anticipate being able to submit a revised manuscript including the new experiments in less than two months. We expect that the cryo-EM experiments can be completed within 2-3 weeks. However, based on our experience with cryo-EM techniques that are not routine and require more than one high-end microscope to be online, we have learned to be conservative when making predictions about our timelines. We are confident that two months will suffice to perform the described experiments, regardless of potential instrument downtime.

- Authors observe finger-like projections associated with migrating membranes. They provide convincing evidence that these projections are associated with the DMP complex and they propose that they result from "tethering of the engulfing membrane to newly synthesized PG via DMP". However, these finger-like projections are still observed at the leading side of the engulfing membrane, although being less prominent, when PG synthesis is inhibited by cephalexin (Figure 4N and subsection “3D architecture of the leading edge of the engulfing membrane”, second paragraph).Do authors really consider that PG synthesis is fully inhibited in the presence of cephalexin? If so, remnant finger-like projections and membrane migration still occurs when PG synthesis is inhibited. They might result from tethering of the engulfing membrane to old PG surrounding the forespore via DMP. In that case, finger-like projections may be stabilized/enhanced (rather than "caused") by tethering of the engulfing membrane to newly synthesized PG. To resolve this problem, the authors need to test the extent of PG synthesis inhibition by Cephalexin.

Thank you for this comment, which made us realize that some of our descriptions were imprecise.

First, we did not completely explain the impact of antibiotic treatment on septal architecture and engulfment. We previously showed that engulfment membrane migration does not continue when PG synthesis is inhibited (see Ojkic et al., 2016, Figure 1 and its supplements), although forespore growth causes the septum to stretch and curve into the mother cell, making it appear that membrane migration has occurred. Measurements of individual cells during timelapse microscopy shows that the distance between the leading edges of the engulfing membrane decreases during engulfment in untreated cells. However, after antibiotic treatment, this distance does not decrease, indicating that the mother cell membrane does not migrate around the forespore. The case of cephalexin-treated cells is more complicated than other antibiotics, because after the septum curves into the mother cell, the leading edge sometimes retracts on one side while advancing slightly on the other. This appears to consist of rotation of the ‘cup’ formed by the bulging septum relative to the lateral cell wall, rather than membrane migration, because the distance between the leading edges does not decrease during this process. Thus, rotation of the septal cup does not increase the degree to which the forespore is engulfed. We are uncertain as to the mechanistic basis for this rotation, but our timelapse experiments indicate that it is a consequence of cephalexin treatment, although it probably also happens after treatment with some penicillins (Ojkic et al. Figure 1—figure supplement 1). Cephalexin inhibits the earliest stages of cell division (Adam et al., 1997; Eberhardt et al., 2003; Kocaoglu et al., 2012), and therefore we speculate that it might be required to tether the extending septum to the lateral cell wall. In the absence of these bridges, the septum might be free to rotate according to Brownian motion, perhaps anchored by the Q-AH ratchet that can also mediate engulfment in the absence of the cell wall (Broder and Pogliano, 2006). We will describe the phenotypes produced by antibiotic treatment more precisely in the text. One area where this will clearly be necessary is where we refer to the side of the membrane that has retracted as the “lagging side” and the other one as the “leading side”, which may have created the misunderstanding that membrane migration happens in cephalexin-treated cells. A revision of this section, and expanded description of the phenotypes, will greatly improve our revised submission.

Second, the reviewers correctly point out that our observations are consistent with the hypothesis that the finger like projections are produced via the tethering of the DMP complex to existing peptidoglycan, since short fingers are observed when PG synthesis is inhibited, and that these fingers are either stabilized or enhanced by binding newly synthesized peptidoglycan, since longer fingers are observed in the presence of ongoing PG synthesis. This is supported by prior studies of the two enzymes that degrade peptidoglycan (D and P), which have shown that these comprise a processive enzyme complex in which P binds and cleaves PG first, allowing D to bind and cleave PG (Morlot and Rudner, 2010). Our cell biological data showed that localization of P to the leading edge of the engulfing membrane is decreased but not completely absent when PG synthesis is inhibited (Ojkic Figure 2F-G and associated supplementary figure), suggesting that P can bind to existing PG, but that high affinity binding requires newly synthesized peptidoglycan. This increased binding of P would likely allow the formation of longer finger like projections. Thus, we thank the reviewers for this suggestion, and will revise the manuscript accordingly.

We agree that it is critical to determine the extent of PG inhibition upon antibiotic treatment. The experiments in this manuscript were based on previously published results of time-lapse microscopy in which we assessed if new division events were blocked after antibiotic treatment, thereby using cell division as a read out for PG synthesis (Ojkic et al., Figure 1—figure supplement 2). These studies showed that cell division was completely blocked by cephalexin, indicating that septal PG synthesis was blocked, and that PG synthesis was inhibited within a few minutes of cephalexin treatment, because cells that were already undergoing septation were unable to complete septum formation. To complement this analysis, we will determine the elongation rate of the vegetative cells that are present in sporulating cultures under our experimental conditions, in order to identify an antibiotic or combination of antibiotics that inhibit both septation and elongation, and we will then collect additional cryo-FIB-ET of cells treated with this antibiotic(s). This will allow us to address the question of the extent of PG synthesis inhibition, and the second essential revision (see below). Our preferred antibiotic candidate is Penicillin V, which has a broad affinity for many *B. subtilis* PBPs (Lakaye et al., 1994; Zhao et al., 1999; Kocaoglu, 2012), and which our preliminary data indicates severely affects both septation and cell elongation.

- Figure 4B shows a reduced rounding of the forespore membrane in the presence of antibiotics, providing evidence that PG is synthesized at the leading edge of the engulfing membrane. Based on this observation and a previous study (Ojkic et al., 2016), authors propose a model (Figure 5) in which PG is exclusively synthesized ahead of the leading edge of the engulfing membrane. However, these data do not exclude the possibility that PG synthesis also happens behind the leading edge of the engulfing membrane, or in other words that PG is resynthesized after degradation by the DMP complex. Indeed, as pointed in the previous point, the fact that membrane migration still happens in the presence of antibiotics again questions whether PG synthesis is fully inhibited. In cells displaying incomplete or asymmetrical engulfment (such as those shown in Figure 2), PG synthesis might have continued for a while and stopped, resulting in septal disks containing newly synthesized PG and leading edges showing impaired PG synthesis. Analysis of septum thickness (as performed in Figure 3B-E) in the presence of antibiotics might shed light on this issue: a thinner septum (but not absent if DMP does not totally degrades septal PG) in the presence of antibiotics would suggest that PG is synthesized behind the DMP complex. If septum thickness is independent of the presence of antibiotics, it suggests that either PG is not synthesized behind the DMP complex or that PG is not fully inhibited in the presence of antibiotics. This uncertainty impacts the hypothesis proposed by the authors because PG synthesis happening behind the DMP complex might explain why a slight and homogenous septal thinning is observed (Figure 3); it might also contribute to the residual PG layer observed in the intermembrane space.

Thank you for raising this important point. As proposed by the reviewers, we have analyzed septal thickness in cephalexin-treated sporangia. As seen in Author response image 1, the septal thickness for untreated and cephalexin-treated wild type sporangia is comparable (17.46 ± 1.17 nm for untreated and 18.72 ± 1.13 nm for cephalexin-treated sporangia). This therefore suggests that either PG is not synthesized behind the DMP complex or that PG is not fully inhibited in the presence of antibiotics.

However, our published results (Ojkic et al., 2016) indicate that septal PG synthesis is completely inhibited in cephalexin treated cells, suggesting that PG synthesis does not happen behind DMP. To provide additional evidence on this important point, we will perform the additional experiments explained above to determine if both elongation and septal PG synthesis is completely inhibited by the antibiotics used in our experiments, and if not, to identify an antibiotic treatment or combinatorial treatment that blocks both processes. We will also collect cryo-electron tomograms of sporangia treated with at least one additional antibiotic (probably Penicillin V or a combination of Penicillin V and cephalexin) and perform a similar analysis to further test this point.